# Neurosymbolic Theory Revision through Predicate Invention

## Abstract

Neurosymbolic AI approaches typically assume perfect and complete symbolic knowledge. This assumption limits their usability, as it is unrealistic in dynamic, real-world environments, particularly in domains that require both structured reasoning and perception. To address this issue, we propose a novel methodology that iteratively revises an initial imperfect logical background theory. Our approach, termed NeTheR, performs a limited number of high-impact modifications to improve the model's performance while maintaining the integrity of the original symbolic structure. Historically, theory revision has been achieved by adding or removing symbolic features to improve logical models. In contrast, NeTheR achieves this by leveraging predicate invention to introduce new neural concept representations, allowing us to learn and use concepts beyond those available in the symbolic data. These high-impact modifications, like the insertion of a new neural concept into a specific part of the model, are identified using a variant of the Sharpe ratio, which measures the potential performance gains. Empirical evaluation shows that NeTheR outperforms its baseline competitors.

## 1 Introduction

Neurosymbolic AI (NeSy) is an emerging subfield within artificial intelligence (Gartner, 2024) that combines the strengths of two traditionally distinct approaches: symbolic reasoning and neural reasoning. Symbolic AI, which emphasizes logic, rules, and human-like reasoning, excels in handling abstract, structured knowledge and can be easily interpreted by humans (De Raedt, 2008; Russell & Norvig, 2020). Neural approaches, on the other hand, have contributed significant advances in pattern recognition, perception, and data-driven learning, especially with unstructured data such as images and text. Such neural approaches do, however, often lack explainability, logical consistency, and the ability to generalize from limited data (Rudin et al., 2021). NeSy aims to bridge these gaps, creating systems that can not only learn from vast amounts of data, but also reason in a structured, interpretable manner. By integrating neural networks' learning capabilities with symbolic reasoning's precision and transparency, NeSy holds the potential to tackle complex problems more effectively.

Despite the promise of NeSy, current systems face a strong limitation: they often operate under the assumption of having perfect symbolic knowledge. This is unrealistic in many real-world scenarios, where the environment is too complex to model perfectly. However, it is also inefficient to discard the knowledge, even if it is imperfect. Especially in low data regimes, one should not have to relearn what one already knows. We address these shortcomings by proposing NeTheR (**Ne**urosymbolic **The**ory **R**evision), an approach to automatically revise imperfect symbolic knowledge through the introduction of neural concepts. This approach combines the advances in neurosymbolic research with the principles of theory revision (Mooney & Shavlik, 2021), a field that focuses on refining knowledge bases such as logical rules or probabilistic networks. NeTheR can refine the existing symbolic structure with new, learned neural concept representations, thus compensating for incomplete knowledge.

As an example task, consider classifying whether a given picture is taken by either Alice or Bob, and depicts either a bird or a fish. Additionally, animals photographed by Alice should be white. The only annotated information is the available set of Boolean encoded categorical variables: Photographer(P), Bones(B), and Swims(S). An expert attempts to solve this task using

the logic formula below, which relies on hollow bones to cover birds and swimming for fish.

$$(P = Alice \lor P = Bob) \land (B = Hollow \lor S = True) \tag{1}$$

This solution is imperfect as it does not cover birds without hollow bones, such as penguins, nor the extra constraint for Alice. Existing theory revision methods would struggle to correct this as they are limited to symbolic data. In contrast, NeTheR introduces neural concepts whose value can also depend on subsymbolic data, like images, thereby increasing expressivity. A general overview of NeTheR is shown in Figure 1, which includes the introduction of two neural concepts to refine the formula. For example, given enough data, one neural concept learns whether an animal is white.

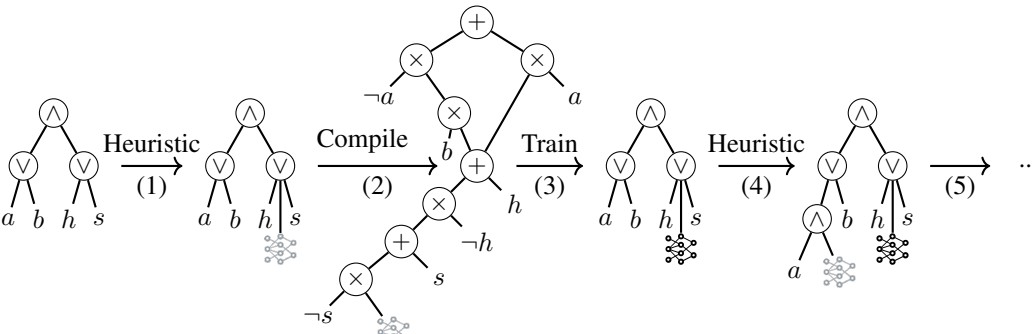

Figure 1: General overview of NeTheR, illustrated by the example. $a$ and $b$ respectively denote whether the picture is taken by Alice or Bob, $h$ denotes whether the animal has hollow bones, and $s$ denotes whether it swims. (1) An untrained neural concept is added; (2) The result is compiled to an arithmetic circuit; (3) This circuit allows us to efficiently compute probabilities and is end-to-end differentiable, such that the neural concept(s) are trained as part of the logical circuit; (4) Once trained, NeTheR performs again a new modification which may add another neural concept; (5) This cycle repeats until the maximum number of modifications is reached or until modifications would decrease performance. An important task is then to efficiently identify high-impact modifications: where do we insert new neural concepts, or what existing concepts should we remove?

Our key contributions are that we introduce NeTheR, which (1) improves neurosymbolic models by refining their imperfect background knowledge in a data-driven manner; and (2) extends theory revision with neural concept representations, thereby allowing models to exploit both symbolic knowledge and subsymbolic data. To achieve this, we furthermore propose an altered version of the Sharpe ratio to identify high-impact refinement modifications.

## 2 BACKGROUND

We first introduce propositional logic as a foundation for formal reasoning, arithmetic circuits for efficient mathematical computation, and neurosymbolic AI, which integrates symbolic reasoning with neural networks.

### 2.1 PROPOSITIONAL LOGIC

A *literal* $l$ is a Boolean variable or its negation ($v$ or $\neg v$). A *propositional logic formula* $F$ is inductively defined as a literal $l$, a disjunction of two formulas $F_1 \lor F_2$ (read as 'or'), a conjunction $F_1 \land F_2$ (read as 'and'), or the negation of a formula $\neg F_1$, with the usual semantics. A *logic formula* acts as a Boolean classifier, evaluating to true or false depending on the values of the Boolean variables. We explicitly support Boolean variables denoting an atom over categorical or continuous variables. For example, $(Colour = Black) \lor (Height < 5)$ is a logic formula over two Boolean variables, respectively denoting an atom over a categorical and continuous variable. A *logic circuit* represents a logic formula as a directed acyclic graph of leaf nodes (literals) and inner nodes with an associated type (either disjunction, conjunction, or negation). While a logic circuit explicitly supports the reuse of subcircuits, i.e., a node can have more than one parent node, we may use logic circuit and logic formula interchangeably throughout this work.

To perform probabilistic inference over a logic formula, the following three structural properties are of interest (Darwiche & Marquis, 2002). A logic formula is in *negation normal form* (NNF) iff negation only occurs directly on the Boolean variables. A formula is *decomposable* iff for every conjunction, the conjunct branches do not share any variables. A formula is *deterministic* iff for every disjunction, the disjunct branches are mutually exclusive, i.e., there is no assignment to the Boolean variables that makes both disjunct branches evaluate to true. To obtain these structural properties (i.e., a d-DNNF formula), various knowledge compilation algorithms have been developed. We will use the PySDD (Meert, 2017) software package for this purpose.

## 2.2 Arithmetic Circuit

If a logic formula $F$ over Boolean variables $X$ is a d-DNNF formula, then we can easily compute the probability of $F$ from the probability of $X$, by replacing each conjunction with multiplication, each disjunction with addition, and each literal with the corresponding probability. This transformation results in an *arithmetic circuit* $a_n$ which can be evaluated for a given instantiation to the Boolean variables $\mathbf{x}$:

$$a_n(\mathbf{x}) = \begin{cases} f_{a,n}(\mathbf{x}) & \text{if node } n \text{ is a literal} \\ \Pi_{d \in in(n)} a_d(\mathbf{x}) & \text{if node } n \text{ is a conjunction} \\ \sum_{d \in in(n)} a_d(\mathbf{x}) & \text{if node } n \text{ is a disjunction} \end{cases}$$

with $f_{a,n}(x)$ a function evaluating an input variable, and negating the result if needed and $in(n)$ the input nodes of node $n$. In our case, $f_{a,n}(x)$ will be a probability value such that $a_n$ represents a probability value as well.

## 2.3 Neurosymbolic AI

The arithmetic circuit described in Section 2.2 evaluates each Boolean variable to obtain its probability, $f_{a,n}(x)$. In a pure inference setting, this evaluates to a fixed value. In a learning setting, it is instead a learnable parameter that is trained from data. Extending this insight, Manhaeve et al. (2021) replaced some of the Boolean variables with the output of a neural network, using a softmax layer to maintain probability values. Because the arithmetic circuit is differentiable, the combination of the circuit and the neural network can be jointly trained end-to-end using gradient descent, which means that we can learn the neural network parameters while exploiting the knowledge represented through a logical circuit. This enables learning through indirect supervision, allows for hard logical guarantees, increases model robustness and interpretability, and improves data efficiency (Manhaeve et al., 2021; Wang & Poon, 2018).

**Neural concepts.** We will similarly extend a logic formula with Boolean variables that are backed by the output of a neural network. This will allow us to introduce new, so-called neural concepts, which are concepts that are not necessarily available in the symbolic data and may only be extractable from subsymbolic data, such as images or time series. Note that this differs from a learned feature as it has no predefined semantic meaning. Inference on this extended formula can be done by compiling the logical circuit into a d-DNNF formula, which can then be converted into an arithmetic circuit. Figure 2 shows the evaluation of such an arithmetic circuit when given the assignment $\{A \mapsto true, B \mapsto false, C \mapsto false\}$ with subsymbolic data feeding into the neural network, by which they evaluate to 0.38 and 0.65. Importantly, we deviate from existing neurosymbolic systems such as DeepProbLog (Manhaeve et al., 2021), where the knowledge is assumed to be specified perfectly, and the neural concepts have a prior intended meaning. In this work, the subsymbolic concept instead obtains meaning during the training process, while converging to a concept that is useful for classification.

We will typically have full observability of the symbolic data, meaning the values $f_{a,n}(x)$ of non-neural backed variables evaluate to 0 or 1 (cf. Figure 2), although this is not required for NeTheR.

## 3 Problem Setting

We consider a binary classification task where symbolic and subsymbolic data are augmented with a possibly imperfect background model that the user may want to revise.

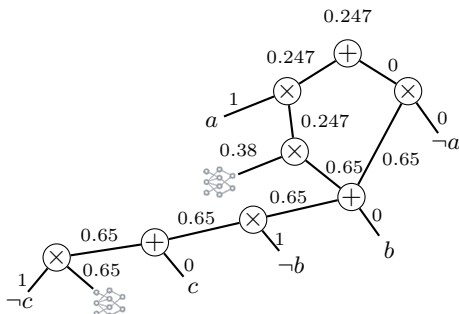

Figure 2: An arithmetic circuit over Boolean input variables and two neural concepts, for a given evaluation $x$. The weight of a neural concept is the output of its neural network for a given input (image). These weights, and those of the symbolic input variables, are denoted above them. The weights propagate upwards, and those results are similarly denoted above the intermediate nodes.

**Binary classification.** Let $X$ be the feature space over values of the symbolic and subsymbolic features, and $Y = \{0, 1\}$ their Boolean label generated by the unknown target model $M^*$. The training set $E$ consists of $m$ instances: $E = \{(x_i, M^*(x_i))|x_i \in X, M^*(x_i) \in Y, i \in [1, m]\}$.
The goal of a binary classification task is to learn a model $M$ that approximates $M^*$, to correctly predict the label of both seen and unseen instances. Given that in practice we do not have access to $M^*$, this is instead achieved through the training set $E$.

**Additional knowledge.** Our setting also comprises an initial, imperfect model $M'$ that serves as additional information to find $M^*$. This addition is motivated by the scenario wherein an expert already has an idea, albeit imperfect, of the target model, and aims to automatically revise it in a data-driven manner. We therefore also include a limit $d_{max}$ on the allowed maximal distance between the input model $M'$ and model $M$, measured using a distance function $d$.

In summary, **given** training set $E$, and an initial model $M'$, we must **find** $M$ such that

$$\arg\max_M F_1(M(X), M^*(X)) \text{ with } d(M, M') \leq d_{max}, \tag{2}$$

with $F_1$ the $F_1$-score used to measure performance in binary classification.

**Theory revision.** We assume that models $M$ and $M'$ are logical circuits over Boolean variables and neural concepts, as described in Section 2. This choice follows from well-known neurosymbolic systems such as DeepProbLog (Manhaeve et al., 2021) and Scallop (Li et al., 2023) that use such circuits during inference and learning. As part of our proposed revision process, we iteratively modify this circuit. A natural choice for $d$ is therefore the minimal number of modification steps to obtain $M$ from $M'$. The types of modifications we consider are discussed in the next section.

## 4 OUR APPROACH: NETHER

We now introduce *NeTheR*, an iterative approach to revise a logic circuit via the incorporation of neural concept representations. The main idea is to iteratively modify the circuit, train the result using indirect supervision, and to repeat until no improvements are found, or until the distance limit $d_{max}$ is reached. To identify the best circuit modification, we use a newly proposed heuristic that estimates the performance impact of each modification. We will first discuss the possible modifications and this selection heuristic, before providing the complete algorithm.

### 4.1 MODIFICATION TYPES

We consider three types of circuit modifications:

1. removing a literal or neural concept $x$, e.g., $(F_1 \vee x) \rightarrow F_1$,

2. replacing a logic circuit node with the conjunction of a newly introduced neural concept $nc$, e.g., $F_1 \rightarrow (F_1 \wedge nc)$,

3. replacing a logic circuit node with the disjunction of a newly introduced neural concept $nc$, e.g., $F_1 \rightarrow (F_1 \vee nc)$.

Note that in case of the second and third modification, the replaced logic circuit node is not necessarily a literal or neural concept, as it can also be an inner node. We do however add an additional restriction in this case, which is to not allow a neural concept to be conjoined or disjoined with another. For example, we do not allow $nc_1 \wedge nc_2$, nor $nc_1 \vee nc_2$, but do allow $(a \vee nc_1) \wedge nc_2$. This leads to the number of possible modifications being linear with the circuit size, as for a circuit with $l$ literals, $n$ neural concepts and $i$ inner nodes, there are $2 * (l + n) + i$ possibilities.

We exclude the insertion of a logical variable as a possible modification type because it is already covered by traditional theory revision methods. In a scenario where the symbolic data is extractable from-, or correlated with, subsymbolic data, this choice does not limit the expressivity. In the opposite scenario, the resulting limitation is resolvable by applying the traditional theory revision methods in conjunction with NeTheR. Even though it is not an explicit modification type, it is possible that the best modification is to perform no modification at all, and that the circuit is considered locally optimal. We elaborate on when this occurs, while discussing the modification selection process.

**Applying a modification.** When inserting a neural concept, we train the underlying neural network in an end-to-end fashion with the rest of the logic circuit, using indirect supervision. For example, if our logic circuit is $M = nc_1 \wedge (a \vee nc_2)$, with $nc$ neural concepts, then we train $M$ using our training dataset; there is no direct label for the neural network. By compiling the logic circuit into d-DNNF prior to training its arithmetic version, we have the benefit of probabilistic semantics (cf. Figures 1 and 2, and Sections 2.2 and 2.3). Integrating a new neural concept enables the logic circuit to capture additional useful features. This is particularly advantageous when the subsymbolic data contains information not yet present in symbolic form, while also increasing robustness against noise in the symbolic data as the inserted neural concept may mitigate inconsistencies.

During training, NeTheR only updates the parameters of the newly inserted neural network and keeps the other parameters frozen. It is however possible to also update the previously trained parameters alongside the newly inserted ones, such that they can be better tuned with respect to each other. To test this hypothesis, we performed an empirical evaluation that is included in Appendix A.1. This showed that not freezing the earlier parameters improved the average $F_1$-score with only 0.02. Therefore, since it also increases the number of trainable parameters and computation time, we keep freezing the previously existing parameters in the rest of our empirical evaluation.

## 4.2 MODIFICATION SELECTION

For the modifications that insert a new neural concept, the impact on $F_1$-score is unknown until we actually train the concept's underlying neural network. This poses a challenge when selecting a circuit modification: since there are several locations where a new concept can be inserted, it is computationally impractical to train each neural network before selecting the best location. To solve this, we will instead estimate the neural network's performance and use a selection heuristic.

**Upper and lower bounds.** To estimate the future performance of a new neural concept prior to training, we can consider all training instances that it covers, which depends on its location within the logic circuit. From these, we can deduce an upper and lower bound on the $F_1$-score. In the best case scenario (B), the new concept corrects all erroneous predictions, while in the worst case scenario (W), it negatively affects all correct predictions. This naturally leads to the first two selection heuristics: selecting the modification with the best best-case (B), or if we are risk-averse, the best worst-case (W) scenario. In case of the former, the result is equivalent to the Bayesian information criterion (Schwarz, 1978) or the Akaike information criterion (Akaike, 1998). In case of the latter, a neural concept is assumed to degrade performance, so NeTheR will only ever remove literals or pre-existing neural concepts, or do nothing.

**Sharpe ratio.** To consider both the best and worst case scenarios simultaneously, we create a more sophisticated heuristic based on the *Sharpe ratio* (Pav, 2021). The Sharpe ratio $S$ is a widely used metric in finance to assess the risk-adjusted performance of an investment $p$. It is derived from the

statistical t-test and is mathematically defined as the difference between $R_p$, which is the expected return of the investment $p$, and $R_f$, which is the return from a theoretical risk-free investment $f$. The denominator $\sigma_p$ is the standard deviation of $p$'s excess returns, quantifying the variability in $p$'s performance. I.e., $S = (R_p - R_f)/\sigma_p$. In essence, this formula highlights how much excess return is achieved per unit of risk taken. A higher Sharpe ratio $S$ indicates a better risk-adjusted performance. As the metric is particularly useful for comparing investments with varying levels of risk, it is frequently used in the context of multi-armed bandit problems (Cassel et al., 2018; Khurshid et al., 2025; Xi et al., 2021).

Similar to the investment $p$, we do not know the performance of a neural concept prior to training and only know its best and worst case impact. In other words, we can use the Sharpe ratio $S$ to objectively trade-off the risk (W) and reward (B) of each circuit modification, and select the best modification as the one with the highest $S$. Using distribution $X_{nc}$ to denote the performance of neural concept $nc$ in a specific location of the logic circuit, its expectation $\mathbb{E}[X_{nc}]$ can serve as a substitution for $R_p$, while the standard deviation $\sigma[X_{nc}]$ replaces $\sigma_p$. Finally, the risk-free rate $R_f$ corresponds to the best modification with a precisely known impact: either the removal of a literal or neural concept, or doing no modification at all.

$$S = \frac{\mathbb{E}[\mathbf{X}_{nc}] - R_f}{\sigma[\mathbf{X}_{nc}]} \tag{3}$$

The distribution $X_{nc}$ depends on both the neural network's architecture and the classification task. In our empirical evaluation, we assume that $X_{nc}$ is a two-point weighted random variable whose mean is $\mu = \alpha B + (1 - \alpha)W$ and variance is $Var(X_{nc}) = \alpha(B - \mu) + (1 - \alpha)(W - \mu)$, and we use these as parameters for a shifted normal distribution. This results in

$$S = \frac{\alpha B + (1 - \alpha)W - R_f}{\sqrt{\alpha((1 - \alpha)B - (1 - \alpha)W)^2 + (1 - \alpha)(\alpha W - \alpha B)^2}} \tag{4}$$

where $\alpha \in (0, 1)$ shifts the distribution closer to the best case when $\alpha > 0.5$. This equation assumes the presence of risk, i.e., $B \neq W$. Risk-free modifications are only selected when the best $S$ is negative, i.e., when the expected $F_1$-score $\mathbb{E}[\mathbf{X}_{nc}]$ is worse than the best risk-free modification (which may be modifying nothing). An example of these calculations is shown in Appendix B.

**Dynamic Sharpe ratio.** Instead of choosing a fixed value for $\alpha$, we will determine it dynamically: when a neural network $nc$ is trained, we adjust $\alpha$ such that the estimated $\mathbb{E}[X_{nc}]$ better matches $nc$'s actual performance. Denoting $\alpha_t$ as the correct value according to the neural network's performance, we move $\alpha$ closer to this value by updating it $\alpha \leftarrow (\alpha + \alpha_t)/2$. In addition to the periodic updates of $\alpha$, we also choose a more informed initial value by randomly selecting $k$ insertion modifications and evaluating the predicted performance with its actual performance. We select a modification with worst case $W$ and best case $B$ and train the network, resulting in $F_1$-score $x$. $\alpha$ is then $\frac{x - W}{B - W}$, showing how close we are to the best case. When $k > 1$, $\alpha$ is the average of all $k$ estimates. By dynamically adjusting $\alpha$, and thereby improving the risk estimation, the heuristic effectively captures performance trends across different performance scenarios. This is corroborated by the empirical evaluation in Appendix A.2, which shows that selecting the modification based on the dynamic Sharpe ratio results in superior performance compared to the alternatives (B, W, and Sharpe ratio with $\alpha = 0.5$).

### 4.3 ALGORITHM

We now discuss the outline of NeTheR using Algorithm 1. First, we determine a suitable, initial $\alpha$ parameter (line 3) to better estimate the potential impact of each circuit modification (line 6 and 7). Then, we iteratively apply the next best modification until the maximum distance is reached (line 5), or until the next best modification is to make no change at all (line 9). Note that it is expensive to compute distance $d(M, M')$, so we approximate it by counting the number of performed modifications, potentially overestimating the true distance. This best modification is greedily selected, out of those that have not yet been tried before, as the one with the highest dynamic Sharpe ratio (line 8). After this selection, we apply the modification (line 10), meaning we modify the logic circuit $M$, recompile it into a d-DNNF representation, and perform end-to-end parameter learning to update the neural network weights. Once the learning process is complete, we can verify whether the estimated

performance was correct, and update $\alpha$ accordingly to improve future estimates (line 11). Finally, if the $F_1$-score improved the best found result, we update $M_{best}$ and continue to further modify the result (line 12 and 13). In case the $F_1$-score did not improve, $M$ resets back to $M_{best}$ (line 13), and the while loop repeats with the next best untried modification, using the best model found so far. Finally, we return the best found model (line 14).

---

**Algorithm 1** NeTheR

---

**Input:**
 1: Imperfect logic circuit $M'$, Maximum distance $d_{max}$,
 2: Training and validation set $T, V \subseteq E$, Number of modifications to initialize $k$
**Output:** Revised model $M$
 3: $\alpha \leftarrow$ initialise_alpha$(M', k)$
 4: $M, M_{best} \leftarrow M'$
 5: **while** $d(M', M) \leq d_{max}$ **do**
 6:     $\mathcal{M}' \leftarrow$ all untried modifications for $M$
 7:     Order $\mathcal{M}'$ based on dynamic Sharpe ratio $S$
 8:     Select best modification $mod$ from $\mathcal{M}'$
 9:     **if** $mod$ is None **then** *break*
 10:     $M \leftarrow apply(M, mod, T)$
 11:     **if** $mod$ adds a neural concept **then** update $\alpha$
 12:     **if** $F_1(M, V) > F_1(M_{best}, V)$ **then** $M_{best} \leftarrow M$
 13:     $M \leftarrow M_{best}$
 14: **return** $M_{best}$

---

## 5 EVALUATION

We empirically evaluate NeTheR on relevant benchmarks, answering the question of whether NeTheR is more effective at solving the revision problem (cf. Section 3) compared to alternative approaches. We investigate two scenarios: a biased scenario where the initial model $M'$ is structurally already very close to the target model $M^*$, and a scenario where it is not. The used code will be made publicly available upon paper acceptance.

### 5.1 EXPERIMENTAL SETUP

We consider 9 open-source benchmark datasets, the details of which are included in Appendix C. Six of those datasets have images as subsymbolic data, whilst three datasets demonstrate our method's ability to also operate on non-image data, such as music and time series.

The 9 open-source datasets do neither include an imperfect initial model $M'$, nor a target model $M^*$ from which we can extract a structurally close $M'$. Hence, we generate for each dataset 24 logic circuits $M^*$, resulting in 216 problem instances in total. From $M^*$ we obtain an initial model $M'$ that is imperfect but structurally close, by systematically removing variables until the $F_1$-score is in the range of $0.6$ to $0.9$. Importantly, we also remove these variables from the training dataset, as they might not be present in the symbolic data in practice. This means NeTheR must revise $M'$ using only the subsymbolic features, and the symbolic features already present in $M'$.

For the experiment without a structural bias, we use as initial model $M'$ a decision tree that is learned for each problem instance. After filtering on imperfect initial models $M'$, i.e., those with an $F_1$-score below $0.9$, we obtain 171 instances. These represent an expert's imprecise knowledge that is not necessarily structurally close to $M^*$. Similar to the experiment with a bias, we restrict the symbolic features to only those present in $M'$. Appendix D shows two instances of the problem for both the structurally biased and unbiased experiment. Appendix E includes an additional, similar, experiment on the derm7pt dataset, wherein we use the existing dataset labels instead of $M^*$.

As our distance function $d$ tracks the number of modifications, we set $d_{max}$ to the number of systematic removals from $M^*$ to $M'$. Each dataset was split into 80% training, 10% validation and 10% test data. For NeTheR, we use $k = 3$ to estimate an initial $\alpha$, and freeze the parameters of previously added neural concepts when training.

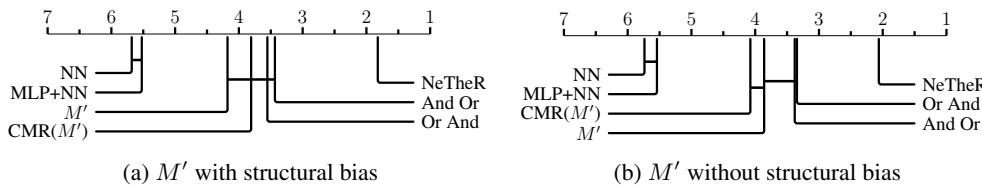

(a) $M'$ with structural bias                    (b) $M'$ without structural bias

Figure 3: Critical difference diagrams (Demšar, 2006; Benavoli et al., 2016), demonstrating the effectiveness of NeTheR. It plots the average rank (closer to 1 is better) and connects with a horizontal line those methods whose performance is not significantly different. We include input model $M'$.

**Used neural networks.** Depending on the type of subsymbolic data, we use a different type of neural network to underlie the neural concepts. For images, we use a pre-trained ResNet-50, with a 6-layer multi-layer perceptron (MLP); for music (FMA dataset), we use an MLP; and for time series (StarLightCurves and Motifs), we use the InceptionTime network (Fawaz et al., 2020). All of these are trained for 50 epochs with patience set to 5.

## 5.2 BASELINE COMPARISON

To evaluate the effectiveness of NeTheR, we compare against the following five baseline methods:

**CMR**: A state of the art concept based model that was initialised with $M'$, and that was allowed to learn 20 additional rules (Debot et al., 2024),
**And Or**: A simple but very general extension of $M'$: $(M' \vee nc_1) \wedge nc_2$, with nc new neural concepts,
**Or And**: A simple but very general extension of $M'$: $(M' \wedge nc_1) \vee nc_2$, with nc new neural concepts,
**NN**: A single neural network,
**MLP+NN**: A multimodal network combining the single neural network with an MLP to also exploit symbolic data (Ahsan et al., 2020; Gessert et al., 2020).

The *NN* approach learns a single neural network, and takes on average 6 minutes. The remaining baselines, *CMR*, *And Or*, *Or And*, and *MLP+NN* each learn two networks, and complete on average in 9, 7, 8, and 12 minutes respectively. In comparison, NeTheR trains an average of 1.5 neural networks during the search phase, and another 2.2 to estimate parameter $\alpha$[1]. Again on average, NeTheR completes the full revision process in approximately 17 minutes: 10.5 minutes are spent estimating an initial $\alpha$, 6 minutes are spent training neural networks during the search phase, and the remaining time is spent enumerating the possible modifications and compiling the revised circuit. In case a sufficient estimate for $\alpha$ is available, the initial step of the algorithm ($\sim 11$ minutes) can be avoided entirely. Information on the used computing infrastructure is found in Appendix F.

Figure 3 contains the critical difference diagrams for both experiments, showing that NeTheR clearly outperforms the other approaches. Interestingly, the multimodal neural network *MLP+NN* does not perform significantly better than the single neural network *NN*. This indicates that the additional complexity does not necessarily translate to better results, because the subsymbolic data may provide a sufficient signal to replace the symbolic features. We also observe that both of these methods frequently perform worse than $M'$. This illustrates well that starting from an existing imperfect model $M'$ has obvious performance benefits. The other three methods, *CMR*, *And Or*, and *Or And*, do exploit the existing information from $M'$. Still, their performance often significantly lags behind NeTheR's. We believe this is because NeTheR is more robust against bad performing neural concepts through a more fine-grained placement. In the other methods neural networks have a central role, heavily influencing the performance: for *And Or* and *Or And* this is obvious, in case of *CMR*, a neural predictor is key in deciding which logical rule determines the prediction. This also explains why NeTheR is more performant in data-poor setting compared to the other methods, as shown in Appendix A.3. More detailed numerical results are included in Table 5 and Table 6, in Appendix G.

To conclude, these results show that NeTheR outperforms its competitors on the revision task in both the structurally biased and unbiased case. Furthermore, it underscores the effectiveness of NeTheR in combining imperfect knowledge $M'$ with subsymbolic information.

---

[1]Even though we use $k=3$ when evaluating NeTheR, some instances have fewer locations to insert a neural network such that the average $k$ is lower.

## 6    RELATED WORK

NeTheR is based on three key features: (1) it leverages an imperfect initial model $M'$ by refining it with (multiple) neural concepts, (2) it introduces neural concepts that are not pre-defined, (3) thereby combining symbolic and subsymbolic data.

**Neurosymbolic AI.**    NeSy systems that integrate probabilistic logic programming with neural networks have gained significant attention for their ability to combine reasoning with perception. Deep-ProbLog (Manhaeve et al., 2021), DeepStochLog (Winters et al., 2022), NeurASP (Yang et al., 2020), and Scallop (Li et al., 2023) are just some examples of systems that enable probabilistic reasoning over the outputs of neural networks. Each of these systems are used under the assumption of perfect symbolic knowledge, where the neural concepts have a predefined intended meaning. In contrast, our contribution does not assume perfect knowledge (key feature 1), and we introduce neural concepts that do not have a pre-envisioned meaning (key feature 2).

**Neural predicate invention.**    Predicate invention systems extend an initial vocabulary with new higher-level predicates that are defined in relation to the existing ones, to improve rule learning. In the neural version, the vocabulary includes neural predicates (Liang et al., 2025; Sha et al., 2024). For example, Sha et al. (2024) explores predicates representing shapes, and other visual features, to learn higher-level concepts in structured visual scenes. Similar to the NeSy systems, these (neural) predicates have a pre-envisioned meaning, and are typically learned before performing predicate invention. In contrast, our work introduces concepts whose meanings are not known a priori (key feature 2) but are instead learned from data alongside the theory, requiring the system to both discover and contextualize their roles without explicit semantic grounding from the user.

**Interpretable concept based models.**    Concept based models (CBM) are machine learning models that use human-interpretable concepts as an intermediate layer in its decision-making process. The best known CBM, named the concept bottleneck model, extracts concepts from raw data using a concept encoder, the results of which are passed to a task predictor that makes the final prediction (Koh et al., 2020). These models have been explored in settings with incomplete or imperfect concepts, aligning with challenges related to imperfect knowledge. Two notable neurosymbolic CBMs that are developed for such settings, DCR (Barbiero et al., 2023) and CMR (Debot et al., 2024), learn logic rules over concepts and utilize neural representations to enhance their accuracy, akin to our notion of neural concepts. Among these, only CMR has the ability to both learn rules and incorporate pre-defined logic rules (key feature 1). This means that, even though CMR was not presented as such, it is a viable solution for our problem setting. However, CMR's use of neural concepts is close to the behavior of *And Or*, while NeTheR allows for a much more refined placement of neural concepts. This explains our empirical observation, that NeTheR is more effective in terms of improving $F_1$-score while maintaining the integrity of the original input.

**Theory revision.**    Research on theory revision combines human-engineered, rule-based knowledge with empirically learned knowledge using symbolic, probabilistic, and neural approaches (Ginsberg et al., 1988; Mooney & Shavlik, 2021). Systems such as DUCE, CLINT and (N)EITHER (Baffes & Mooney, 1993; De Raedt, 1992; Muggleton, 1987; Ourston & Mooney, 1990) refined propositional rule bases using abductive reasoning and inductive learning, while FOCL, FORTE and MIS (Kókai et al., 1997; Pazzani & Brunk, 1991; Richards & Mooney, 1991) tackled the revision of first-order Horn clause theories by leveraging inductive logic programming. In contrast to NeTheR, these systems are not fit to capture concepts available in subsymbolic data (key feature 3).

## 7    DICUSSION

In this work, we explore the revision of propositional theories by augmenting them with neural concepts. We characterize the trade-off this introduces between interpretability and predictive performance and identify the methodological advances required to generalize the approach to first-order theories.

**Interpretability of neural concepts.**    Neurosymbolic AI methods exploit existing knowledge to increase accuracy, robustness, with less data. Interpretability also increases compared to purely

neural models, because it enables the use of symbolic information and makes their combination symbolically understandable (i.e., and, or, negation). E.g., "swims=1 $\wedge$ *neural_concept*" is more interpretable than only "neural_concept". NeTheR introduces neural concepts that do not have a prior intended meaning (line 150). This trades-off interpretability for accuracy compared to a fully symbolic initial model $M'$. However, not having a prior intended meaning has only limited impact on interpretability in the context of neurosymbolic AI: a neural network does not necessarily align with an intended concept such that an analysis (for example using saliency maps etc.) is required for a robust explanation regardless of having an intended meaning.

**Future work: generalizing to first-order.** First-order theories are out of scope for this work, but an important aspect to consider is the type of modifications that are explored. In traditional systems like FORTE Richards & Mooney (1991), there are two types of modifications: (1) changing existing rules by removing antecedents or through inverse resolution and (2) adding antecedents/rules through predicate invention. The first type should not be a problem to expand with neural concepts. The second type however increases the search space significantly when compared with NeTHeR, so which constraints we can apply on the possible neural concepts requires more research. For example, (1) would you allow variables to become part of the neural concept and how would you decide these; (2) Is it performant to only add neural concepts to existing rules, thus not allowing new rules to be made? Depending on the answers, the search space may become much larger and new heuristics have to be developed to approximate the performance without training each option.

## 8 CONCLUSION

We introduced NeTheR, an approach for revising imperfect propositional theories that introduces neurally learned concept representations, leveraging neural predicate invention. An altered version of the Sharpe ratio guides the revision, identifying and selecting the modification with the highest impact. The covered problem setting is inspired by earlier work on theory revision, but more suited to modern research fields such as NeSy by combining subsymbolic data with categorical symbolic data. Empirical evaluation on open-source data shows that using the Sharpe ratio to dynamically adjust to the performance of the neural concepts performs better than using a static assumption. When compared to related solutions, NeTheR significantly outperforms, getting an average increase in $F_1$-score of 13%.

## 9 REPRODUCIBILITY

The code for NeTheR and for the experimental evaluation will be released upon paper acceptance. The experimental setup is detailed in Section 5.1, and all evaluations are conducted on open-source datasets, with preprocessing steps described in Appendix C. The proposed algorithm is presented in Section 4.3, and the results of ablation studies are provided in Appendix A.

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

# A  ABLATION STUDIES

The ablation studies were conducted using the experimental setup with a structural bias.

## A.1  FREEZING EXISTING NEURAL CONCEPTS

When adding a neural concept, the underlying neural network is trained using indirect supervision while freezing the parameters of the previously trained neural networks. Table 1 shows that this choice, of freezing previously learned parameters, only slightly decreases performance. In addition, it enhances computational efficiency, as the number of trainable parameters is reduced in subsequent iterations, thereby decreasing the computational load and memory requirements for optimization.

Table 1: Results of ablation study 2, which studies the effect of freezing existing learned neural concepts. The results show only a small decrease in $F_1$-score when freezing previously added neural concepts. $\#NC \geq 2$ is the number of times there were two or more neural concepts.

| Datasets | $F_1$ (Freeze) | $F_1$ (No Freeze) | $\#NC \geq 2$ |
|---|---|---|---|
| ATH | 0.67±0.0 | 0.69±0.0 | 1 |
| AWA2 | 0.90±0.1 | 0.92±0.0 | 11 |
| CelebA | 0.96±0.0 | 0.96±0.0 | 3 |
| CUBB | 0.99±0.0 | 0.99±0.0 | 1 |
| derm7pt | 0.79±0.1 | 0.79±0.1 | 2 |
| FMA | 0.74±0.0 | 0.78±0.0 | 1 |
| StarLightCurves | 0.97±0.0 | 0.98±0.0 | 2 |
| VOC-Pascal | 0.89±0.1 | 0.87±0.1 | 6 |

## A.2  EFFECTIVENESS OF THE SELECTION HEURISTIC

This experiment studies the effectiveness of dynamically updating $\alpha$, compared to the static alternatives as described in Section 4.2. The results, shown in Figure 4 and Table 2, demonstrate that our proposed variant of the Sharpe ratio with a dynamic $\alpha$ outperforms the alternative approaches. By dynamically adjusting $\alpha$, and thereby updating the variance and expected reward of the neural concepts, the Sharpe ratio based heuristic effectively captures performance trends across different performance scenarios.

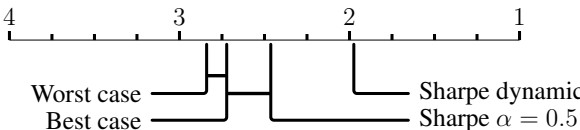

Figure 4: Critical difference diagram, identifying the dynamic Sharpe ratio as the best heuristic. Closer to 1 is better.

## A.3  DATA-EFFICIENCY

We evaluate the data-efficiency by varying the size of the training dataset, for all datasets. Based on the previous performance results, we excluded *And Or* and *Or And* from the comparison because they are worse versions of NeTheR. The results of the data-efficiency experiment are shown in Figure 5 and Table 3. Both NeTheR and *CMR* perform much better in low data regimes because they can rely on $M'$. Compared to *CMR*, NeTheR shows slightly better but similar performance.

Table 2: $F_1$-score results of ablation study 1, which studies the used selection heuristic. The dynamic Sharpe ratio outperforms the static approaches.

| Datasets | Sharpe dynamic | Worst case | Sharpe $\alpha = 0.5$ | Best case |
|---|---|---|---|---|
| ATH | 0.85±0.13 | 0.84±0.13 | 0.85±0.13 | 0.78±0.17 |
| AWA2 | 0.91±0.07 | 0.80±0.11 | 0.84±0.11 | 0.91±0.05 |
| CelebA | 0.90±0.08 | 0.86±0.11 | 0.88±0.11 | 0.86±0.10 |
| CUBB | 0.89±0.09 | 0.86±0.10 | 0.89±0.09 | 0.83±0.10 |
| derm7pt | 0.84±0.13 | 0.82±0.14 | 0.83±0.14 | 0.78±0.16 |
| FMA | 0.83±0.12 | 0.78±0.12 | 0.81±0.12 | 0.81±0.10 |
| Motifs | 0.81±0.14 | 0.79±0.14 | 0.79±0.14 | 0.80±0.13 |
| StarLightCurves | 0.90±0.09 | 0.81±0.09 | 0.83±0.11 | 0.89±0.08 |
| VOC-Pascal | 0.93±0.06 | 0.84±0.12 | 0.88±0.11 | 0.92±0.06 |

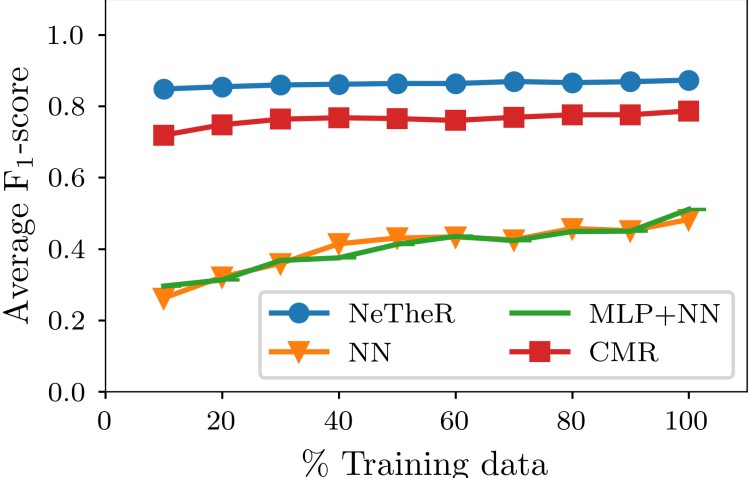

Figure 5: NeTheR is robust to limited training data.

Table 3: Numerical results showing the average $F_1$-score over all datasets per fraction (%) of available training data. It is clear that both NeTheR and CMR are robust against limited training data.

| Methods | $F_1$-score per fraction of available data | | | | | | | | | |
|---|---|---|---|---|---|---|---|---|---|---|
| | 10% | 20% | 30% | 40% | 50% | 60% | 70% | 80% | 90% | 100% |
| NeTheR | 0.85 | 0.85 | 0.86 | 0.86 | 0.86 | 0.86 | 0.87 | 0.87 | 0.87 | 0.88 |
| CMR | 0.72 | 0.75 | 0.76 | 0.77 | 0.77 | 0.77 | 0.77 | 0.78 | 0.78 | 0.79 |
| NN | 0.26 | 0.32 | 0.36 | 0.41 | 0.43 | 0.43 | 0.43 | 0.46 | 0.46 | 0.48 |
| MLP+NN | 0.30 | 0.31 | 0.37 | 0.37 | 0.41 | 0.44 | 0.44 | 0.45 | 0.45 | 0.51 |

## B MODIFICATION SELECTION EXAMPLE

To clarify how the Sharpe ratio is used to select the next modification, consider the following toy example. This is based on the circuit used in the introduction (Figure 1).

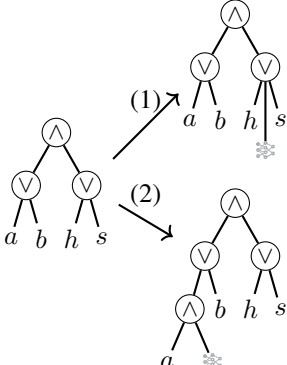

| ID | a | b | h | s | Label |
|----|---|---|---|---|-------|
| 1 | ✓ | ✓ | ✓ | ✓ | ✓ |
| 2 | ✓ | ✗ | ✓ | ✗ | ✓ |
| 3 | ✗ | ✓ | ✗ | ✓ | ✓ |
| 4 | ✓ | ✓ | ✗ | ✗ | ✓ |
| 5 | ✓ | ✗ | ✗ | ✗ | ✓ |
| 6 | ✓ | ✗ | ✗ | ✓ | ✗ |
| 7 | ✗ | ✓ | ✗ | ✗ | ✓ |
| 8 | ✓ | ✗ | ✓ | ✗ | ✗ |
| 9 | ✗ | ✓ | ✗ | ✗ | ✗ |
| 10 | ✗ | ✗ | ✗ | ✗ | ✗ |

The initial circuit covers instances $\{1, 2, 3, 6, 8\}$, thus having 3 true positives : $\{1, 2, 3\}$, 2 false positives :$\{6, 8\}$, 3 false negatives: $\{4, 5, 7\}$ and 2 true negatives:$\{9, 10\}$. This leads to the initial $F_1 = \frac{2 \times TP}{2 \times TP + FP + FN} = \frac{6}{11} \approx 0.55$. We consider in this example two possible modifications:

1. adding a neural concept in the existing disjunction with literals $h$ and $s$
2. adding a neural concept in conjunction with literal $a$.

To estimate which modification will be applied, the Sharpe ratio is calculated based on the best and worst case performance of each neural concept and the best risk-free modification.

**Best and worst cases.** The best case for modification 1 is that instances $\{4, 5, 7\}$ are classified as positive by the neural concept, thus converting 3 false negatives into true positives, giving $B_1 = \frac{12}{12+2+0} = \frac{12}{14} \approx 0.86$. The worst case is that instance 9 is classified as positive, thus converting one true negative into a false positive, giving $W_1 = \frac{6}{6+3+3} = \frac{6}{12} = 0.5$. Note that instance 10 is not affected by the modification, as both $a$ and $b$ are false, thus making circuit predict false independent of the neural concept. For the second modification, the best case is that instances $\{6, 8\}$ are classified as negative by the neural net, converting 2 false positives into true negatives, giving $B_2 = \frac{6}{6+0+3} = \frac{6}{9} \approx 0.66$. The worst case is that it classifies instance 2 as negative, converting a true positive into a false negative, giving $W_2 = \frac{4}{4+2+4} = \frac{4}{10} = 0.4$.

**Risk-free modification.** To calculate the risk-free rate for the Sharpe ratio, we measure the performance of all removal operations:

Removing $a$ from the circuit results in $F_1 = \frac{4}{4+0+4} = \frac{4}{8} = 0.5$

Removing $b$ from the circuit results in $F_1 = \frac{4}{4+2+4} = \frac{4}{10} = 0.4$

Removing $h$ from the circuit results in $F_1 = \frac{4}{4+1+4} = \frac{4}{9} \approx 0.44$

Removing $s$ from the circuit results in $F_1 = \frac{4}{4+1+4} = \frac{4}{9} \approx 0.44$

Performing no modification: $f_1 = 0.55$

The best risk-free modification $R_f = 0.55$ is thus doing no modification at all.

**Calculating the Sharpe ratio.** Say that we do not have prior information over the performance of the neural network, thus setting $\alpha = 0.5$. The Sharpe ratios of both modifications are then calculated as shown in equation (4): $S_1 = \frac{\alpha B_1 + (1-\alpha)W_1 - R_f}{\sqrt{\alpha((1-\alpha)B_1 - (1-\alpha)W_1)^2 + (1-\alpha)(\alpha W_1 - \alpha B_1)^2}} = 0.72$. In similar fashion, we calculate $S_2 = -0.17$, thus showing that applying the first modification would result in the best estimated performance gain.

# C EXPERIMENT DETAILS

We consider multiple benchmark datasets, the details of which we list below. We also clarify the generation of ground truth theories $M^*$ and imperfect theories $M'$.

## C.1 BENCHMARK DATASETS

**Austin Texas Housing (ATH) (Pierce, 2021).** All images were resized to 64×64 pixels, and irrelevant attributes (zpid, description, streetAddress, zipcode, latitude, longitude, latest_saledate) were removed to streamline the dataset for the task.

**AWA2 (Xian et al., 2019).** All images were resized to 64×64 pixels and used as subsymbolic data. The provided predicates associated with each class were utilized as symbolic data.

**CelebA (Liu et al., 2015).** We used the first 10,000 images in this dataset, all resized to 64x64 pixels. The attributes related to each picture were used as symbolic data.

**CUBB (Wah et al., 2011).** All images were resized to 64×64 pixels and used as subsymbolic data. The provided attributes associated with each picture that had a certainty greater than 1 were utilized as symbolic data.

**derm7pt (Kawahara et al., 2019).** The clinical images were resized to 64x64 pixels and used as subsymbolic data. The given metadata and 7-point structure attributes were used as symbolic data.

**FMA (Defferrard et al., 2017).** For this dataset, we used the 'large' subset and discretized both the track data and Echonest data to obtain symbolic representations. The provided audio features were retained as subsymbolic data for analysis.

**Motifs.** This dataset comprises time series that were generated using TSMD-Bench (Van Wesenbeeck et al., 2024), constructed from the six core motifs from the NATOPS time series (Song et al., 2011). We extended these six core motifs to 18, by shifting their amplitude with +2 or -2. This simulates a common occurrence, for example, leakages cause a fixed amplitude to existing patterns. This is a challenging component in time series classification tasks, as instances have to be differentiated in both amplitude and shape. Then, using these 18 motifs, we generated 10,000 random instances by concatenating 10 random motifs per instance. Each instance thus comprises 180 Boolean values as symbolic data, indicating whether a specific motif appears in a particular position. Figure 6 illustrates half of an instance with its symbolic representation.

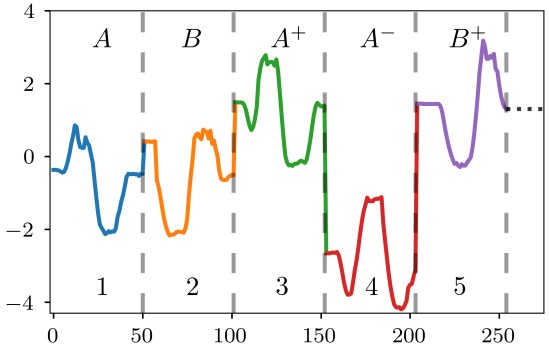

Figure 6: Visualization of five windows of one instance in the motifs dataset. $X_i^+$ and $X_i^-$ both denote motif $X$ in window $i$, with respectively a positive and negative shift in amplitude. The symbolic data of this instance would be $\{A_1=\top, B_2=\top, A_3^+=\top, A_4^-=\top, B_5^+=\top\}$, with the remaining variables $\bot$.

**StarLightCurves (Rebbapragada et al., 2009).** Each instance in this time series dataset was divided into three equal windows. Within each window, the following features were identified: large peak, deep trough, low variability, high variability, sinusoidal shape, high average and large difference between min and max. Such features can be computed automatically and are often used in deployed systems (Christ et al., 2018). Each feature-window combination was represented as a Boolean variable to mimic expert detection of anomalies or behaviours based on patterns in specific parts of the time series.

**VOC-Pascal.** All images were resized to 64×64 pixels and used as subsymbolic data. We used the presence of an object in the image as symbolic data.

## C.2 Generating $M^*$ and $M'$.

For each dataset, we generated 24 logic circuits $M^*$ and corresponding imperfect circuits $M'$. The construction process for $M^*$ is top-down, starting from a disjunctive root node and iteratively refining the lower nodes. Each branch of the root disjunction contains nested conjunction nodes, and the leaf nodes are atomic variable comparisons using categorical or continuous columns from the data (e.g., $age < 30$ or $temperature = High$). Each node is generated with a degree of randomness, becoming either a nested subtree or a leaf node. In case of the latter, a random valid option (attribute, comparison operator and value) is chosen based on the dataset. The structurally biased imperfect theory $M'$ is obtained from $M^*$ by selecting a set of random attributes that are present within $M^*$, and by removing all leaf nodes using those variables. If the resulting $M'$ has an $F_1$-score that is not between 0.6 and 0.9, the process for generating an $M^*$ and corresponding $M'$ starts over. This resulted in $M^*$ circuits that range from 3 to 47 nodes with a mean and standard deviation of $14 \pm 9$ nodes, with corresponding $M'$ circuits ranging from 1 to 39 nodes and a mean and standard deviation of $8 \pm 8$ nodes. To learn the decision trees for the experiment without structural bias, we use the training data and alter the parameters of the sklearn decision tree learner until the tree has an $F_1$-score smaller than 0.9.

# D Qualitative examples

Figures 7 and 8 show qualitative examples of the empirical evaluation, illustrating the ground truth theory $M^*$, the input theory $M'$, and the revised model on two of the used benchmark datasets.

# E Experiment On Existing Labels

The experiments in Section 5 are conducted on the generated target model $M^*$. In this section we again repeat the non-biased experiment, but now using the existing labels of the dataset. We focus on derm7pt as it is a binary classification task, inferring the presence of melanoma. Instead of using the multitask approach that is proposed in the original paper Kawahara et al. (2019), we learned a classifier (ResNet-50) for each of the 7-point structures using both the dermoscopic and clinical images. After that, we use the predicted values for the 7-point structures to construct the 7-pt score, which is combined with the metadata to serve as the data to learn a decision tree on. This decision tree can then be revised using NeTheR with both the dermoscopic and clinical images as subsymbolic data. We include as reference the models that were specifically designed for this task, and that were trained on unbalanced datasets: the 7pt-Unbalanced and Direct-unbalanced results that can be found in the benchmark paper (Kawahara et al., 2019). Table 4 summarizes the results, showing that the learned decision tree already outperforms the originally proposed approaches, and that NeTheR further improves performance.

# F Hardware

The experiments are performed on a machine running Ubuntu 20.04.6 LTS with 128 GB memory, an Intel(R) Xeon(R) CPU E5-2630 v4 @ 2.20GHz CPU and a NVIDIA GeForce GTX 1080 Ti GPU.

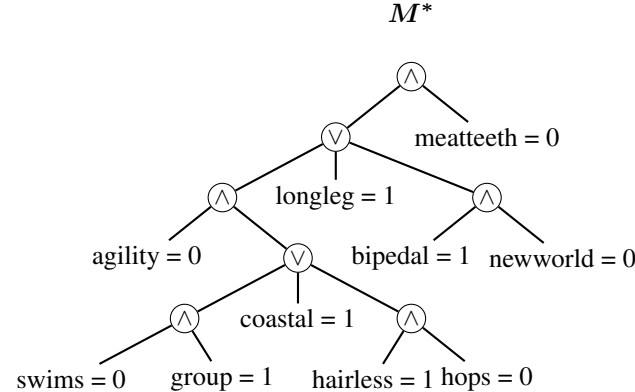

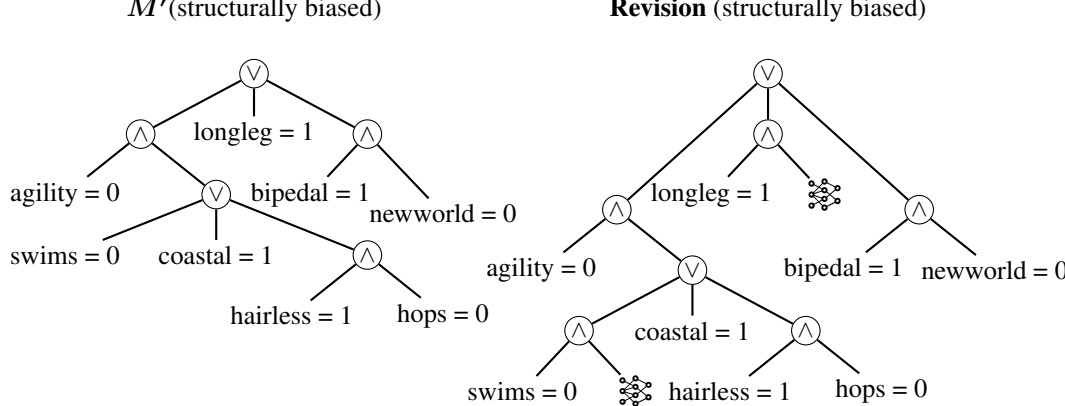

Figure 7: An example of an instance of the AWA2 dataset. The $F_1$-score for the structurally biased experiment increased from $0.77$ for $M'$ to $0.81$ for the revised model of NeTheR. For the unbiased experiment, the learned decision tree reached an $F_1$-score of $0.78$, while its revision using NeTheR outperforms with an $F_1$-score of $0.93$.

## G  DETAILED EXPERIMENTAL RESULTS

Table 5 and 6 summarize the numerical results of the biased and non-biased experiment respectively, by providing for each dataset the mean improvement of $F_1$-score over the initial model $M'$. The $F_1$-score for $M'$ itself is provided in Table 7.

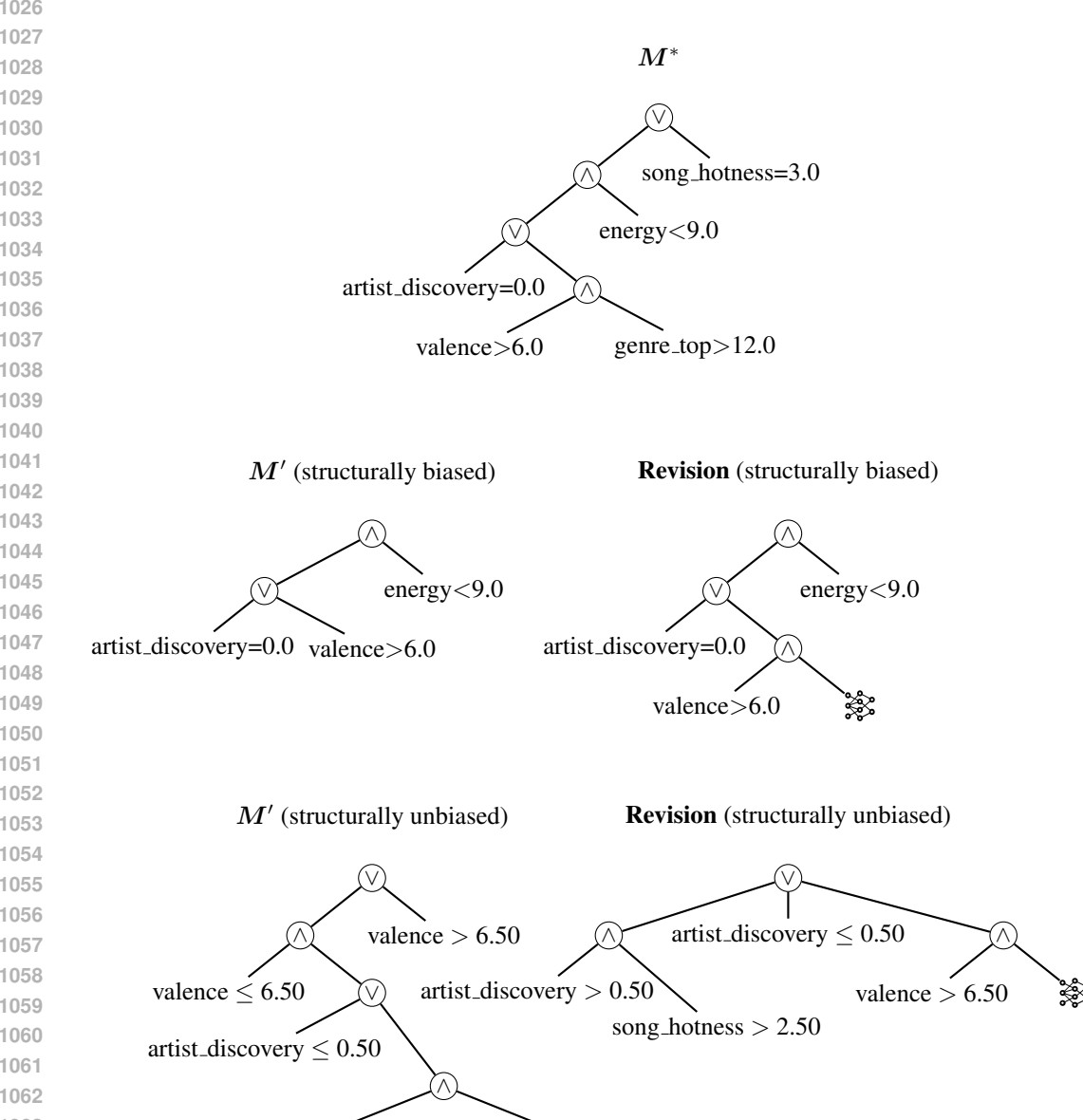

Figure 8: An example of an instance of the FMA dataset. The $F_1$-score for the structurally biased experiment increased from $0.71$ for $M'$ to $0.89$ for the revised model of NeTheR. For the unbiased experiment, the learned decision tree reached an $F_1$-score of $0.67$, whilst its revision using NeTheR outperforms with an $F_1$-score of $0.85$.

Table 4: AUROC scores on the melanoma inferring task of the derm7pt dataset.

| Methods | AUROC-score |
|---|---|
| Kawahara et al. (7pt-Unbalanced) | 76.8 |
| Kawahara et al. (Direct-Unbalanced) | 83.2 |
| Learned Decision Tree (M') | 85.0 |
| NeTheR | 86.9 |

Table 5: Comparison with other methods, demonstrating the effectiveness of NeTheR when $M'$ is structurally biased. The values are the mean differences in $F_1$-score between the learned model $M$ and the input model $M'$, i.e., $F_1(M(x), M^*(x)) - F_1(M'(x), M^*(x))$, aggregated per dataset. Higher is better. The $F_1$-scores for the input model $M'$ are shown in Table 7.

| Datasets | Methods | | | | | |
|---|---|---|---|---|---|---|
| | **NeTheR** | And Or | Or And | NN | MLP+NN | CMR |
| ATH | **+0.09±0.1** | -0.13±0.2 | -0.15±0.2 | -0.38±0.3 | -0.32±0.3 | +0.00±0.1 |
| AWA2 | **+0.14±0.1** | +0.13±0.1 | +0.12±0.1 | -0.03±0.1 | -0.07±0.1 | +0.10±0.1 |
| CelebA | **+0.12±0.1** | +0.04±0.1 | +0.04±0.1 | -0.15±0.2 | -0.25±0.3 | +0.00±0.1 |
| CUBB | **+0.10±0.1** | -0.02±0.1 | -0.02±0.1 | -0.32±0.3 | -0.38±0.4 | -0.01±0.1 |
| derm7pt | **+0.07±0.1** | -0.26±0.3 | -0.31±0.3 | -0.59±0.3 | -0.56±0.3 | -0.06±0.1 |
| FMA | **+0.09±0.1** | +0.04±0.1 | +0.04±0.1 | -0.25±0.3 | -0.05±0.2 | -0.05±0.1 |
| motifs | **+0.06±0.1** | -0.09±0.2 | -0.13±0.3 | -0.64±0.2 | -0.63±0.2 | -0.06±0.2 |
| StarLightCurves | **+0.13±0.1** | -0.15±0.4 | +0.04±0.3 | -0.25±0.4 | -0.10±0.4 | +0.12±0.1 |
| VOC-Pascal | **+0.19±0.1** | +0.17±0.1 | +0.17±0.1 | +0.07±0.2 | +0.07±0.2 | +0.14±0.1 |

Table 6: Comparison with other methods, demonstrating the effectiveness of NeTheR when $M'$ is not structurally biased. The values are the mean differences in $F_1$-score between the learned model $M$ and the input decision tree $M'$, i.e., $F_1(M(x), M^*(x)) - F_1(M'(x), M^*(x))$, aggregated per dataset. Higher is better. The $F_1$-scores for the input model $M'$ are shown in Table 7.

| Datasets | Methods | | | | | |
|---|---|---|---|---|---|---|
| | **NeTheR** | And Or | Or And | NN | MLP+NN | CMR |
| ATH | **+0.06±0.0** | -0.08±0.2 | -0.03±0.1 | -0.53±0.3 | -0.38±0.2 | -0.28±0.2 |
| AWA2 | **+0.07±0.1** | +0.02±0.1 | +0.02±0.1 | -0.09±0.1 | -0.13±0.1 | +0.06±0.1 |
| CelebA | **+0.07±0.1** | +0.05±0.1 | +0.04±0.1 | -0.14±0.2 | -0.25±0.3 | -0.05±0.2 |
| CUBB | **+0.07±0.1** | -0.02±0.1 | +0.02±0.1 | -0.30±0.3 | -0.36±0.4 | -0.23±0.2 |
| derm7pt | **+0.03±0.1** | -0.02±0.1 | -0.02±0.1 | -0.65±0.3 | -0.64±0.3 | -0.14±0.1 |
| FMA | **+0.06±0.1** | +0.01±0.1 | +0.01±0.1 | -0.35±0.3 | -0.13±0.3 | -0.05±0.1 |
| motifs | **+0.13±0.2** | -0.19±0.4 | -0.18±0.4 | -0.56±0.2 | -0.54±0.3 | +0.00±0.3 |
| StarLightCurves | +0.11±0.1 | +0.06±0.2 | +0.01±0.3 | -0.21±0.3 | -0.12±0.3 | **+0.12±0.1** |
| VOC-Pascal | +0.15±0.1 | +0.11±0.1 | +0.10±0.1 | +0.07±0.2 | +0.07±0.2 | **+0.16±0.1** |

Table 7: Average $F_1$-score of input model $M'$ for each dataset. $M'_1$ is the model used for the structurally biased experiment, while $M'_2$ is the learned imperfect decision tree that is used for the non-biased experiment.

| Dataset | $F_1(M'_1(x), M^*(x))$ | $F_1(M'_2(x), M^*(x))$ |
|---|---|---|
| ATH | 0.76±0.1 | 0.80±0.1 |
| AWA2 | 0.77±0.1 | 0.84±0.1 |
| CelebA | 0.78±0.1 | 0.74±0.1 |
| CUBB | 0.80±0.1 | 0.78±0.2 |
| derm7pt | 0.77±0.1 | 0.81±0.1 |
| FMA | 0.74±0.1 | 0.78±0.1 |
| motifs | 0.75±0.1 | 0.64±0.2 |
| StarLightCurves | 0.77±0.1 | 0.75±0.1 |
| VOC-Pascal | 0.74±0.1 | 0.70±0.2 |

