# OpenReview forum: "Neurosymbolic Theory Revision through Predicate Invention"
_ICLR.cc/2026/Conference — Submitted to ICLR 2026_

### Official Review · Reviewer_KUoG · 2025-10-26

**Soundness:** 3
**Presentation:** 3
**Contribution:** 3
**Rating:** 6
**Confidence:** 3

**Summary:**

This paper proposes NeTheR (Neurosymbolic Theory Revision), which revises an imperfect propositional theory by inventing neural predicates (“neural concepts”) and making a small number of high-impact edits to an initial logic circuit M’. Candidate edits include removing literals/concepts or replacing a circuit node with a conjunction/disjunction that inserts a new neural concept. After each edit, the circuit is compiled to d-DNNF and converted to an arithmetic circuit, enabling end-to-end training of the inserted concept via backpropagation through the circuit. A dynamic Sharpe-ratio heuristic ranks edits using bounds on best/worst-case F1 improvement, with a data-driven update of \alpha to balance risk/reward. Experiments across nine datasets (images, time series, music) show consistent gains over baselines in both biased (initial M’ close to target) and unbiased (decision-tree M’) regimes, with ablations for the selection heuristic and concept-freezing choice.

**Strengths:**

- Originality (neural predicate invention for theory revision). Combines classic theory revision with modern NeSy by learning and inserting neural concepts at targeted locations, rather than only tuning predefined concepts; the dynamic Sharpe-ratio edit selection is a simple, risk-aware idea that works well in practice.
- Quality (clear algorithm + solid evaluation). The pipeline is explicit (Algorithm 1), uses d-DNNF to enable differentiable arithmetic circuits, and includes ablations (dynamic vs. static Sharpe; freeze vs. no-freeze), critical-difference plots, and timing breakdowns.
- Clarity (figures & examples). Fig. 1 illustrates the “insert–compile–train–rescore” loop; Fig. 2 clarifies arithmetic-circuit evaluation with neural concepts; appendices detail datasets and protocols.
- Significance (data-efficient revisions). NeTheR improves over M’ and concept-based baselines in both biased/unbiased regimes and is robust in low-data settings; a derm7pt study with real labels shows additional promise.

**Weaknesses:**

- Ecological validity of M’/targets. Main setups synthesize M^* and derive M’ by variable removal; more human-authored imperfect theories (beyond derm7pt) would strengthen real-world relevance.
- Scalability/complexity. Each edit requires d-DNNF compilation and (often) training; search enumerates O(|\text{circuit}|) placements. Provide scaling curves/failure cases for larger circuits and feature vocabularies.
- Heuristic grounding. Dynamic Sharpe is effective but lacks theoretical guarantees; analyze sensitivity to \alpha, compare with alternative acquisition rules (e.g., UCB/Thompson), and discuss robustness to mis-specification.
- Expressivity & scope. Propositional setting and the restriction against direct nc∧nc / nc∨nc insertions may limit interactions; discuss paths to relational theories and richer compositions.

**Questions:**

1. Construction/generalization of M’. Beyond derm7pt, can you include more expert-written imperfect theories? How does NeTheR handle structural mismatch (wrong connectives) vs. predicate omission?
1. Search/compile scaling. What are typical candidate-edit counts and compilation/training times as circuit size grows? Any compile failures or memory limits? A size-vs-latency plot would help.
1. Heuristic details. How is \alpha_t derived from trained-concept performance, and how often is \alpha updated before convergence? Results for alternative acquisition functions?
1. Concept freezing. Would a final joint fine-tuning pass of all learned concepts offer further gains at reasonable cost?
1. Interpretability. Do invented concepts admit post-hoc grounding (saliency/prototypes/symbolic correlates) on real datasets? Provide qualitative examples.
1. Expressivity/relational lift. What is required to support first-order theories (lifted compilation, predicate sharing) while keeping search tractable?

---

> ### Author Response · Authors · 2025-11-21
>
> >Beyond derm7pt, can you include more expert-written imperfect theories? How does NeTheR handle structural mismatch (wrong connectives) vs. predicate omission?
>
> On derm7pt we use as $M'$ a decision tree learned on the original labels instead of an expert-written theory. The structurally unbiased experiment uses learned decision trees as $M'$. The results show that despite the structural mismatch, the approach still works. An example of this mismatch is shown in appendix D, where we added 2 examples of $M*$, the structurally biased and unbiased $M'$, and its revisions.
>
> >What are typical candidate-edit counts and compilation/training times as circuit size grows? Any compile failures or memory limits?
>
> The number of possible modifications ranges from 2 to 61 possibilities, calculated as shown in paragraph 4.1. There were no memory or compile failures. Growing the circuit size has marginal effect on the compilation and training times, hence why we did not include these plots. The biggest difference is seen in the time to construct all modifications.
>
> >How is $\alpha_t$ derived from trained-concept performance, and how often is $\alpha$ updated before convergence? Results for UCB/Thompson acquisition?
>
> $\alpha$ is derived as follows: we select a modification with worst case $W$ and best case $B$ and train the network, resulting in $F_1$-score $x$. $\alpha$ is then $\frac{x-W}{B-W}$, showing how close we are to the best case. When $k>1$, $\alpha$ is the average
>     of all $k$ estimates. We have clarified this in the text (lines 305-307).
>
> Standard UCB and Thompson are not relevant in our setting for two reasons. (1) the non-stationary setting: performing a modification influences the reward of a following modification. (2) once a modification is performed we know its effect, a modification need not be done multiple times to refine the estimate of its effect.
>
> >Would a final joint fine-tuning pass of all learned concepts offer further gains at reasonable cost?
>
> The gain from not freezing the multiple concepts was marginal in our empirical evaluation (Table 1). If there is no time constraint, we expect this to improve the results slightly.
>
> >Do invented concepts admit post-hoc grounding (saliency / prototypes / symbolic correlates) on real datasets?
>
> Please see our shared response. If this does not answer your question, feel free to let us know.
>
> >What is required to support first-order theories (lifted compilation, predicate sharing) while keeping search tractable?
>
> First-order theories are out of scope for this work, but an important aspect to consider would be the type of modifications that are explored. In traditional systems like FORTE, there are two types of modifications:
>
>  (1) changing existing rules by removing antecedents or through inverse resolution and
>
>  (2) adding antecedents/rules through predicate invention.
>
> The first type should not be a problem to expand with neural concepts. The second type however increases the search space compared to NeTHeR, so which constraints we can apply on the possible neural concepts requires more research.  For example, (1) would you allow variables to become part of the neural concept and how would you decide these; (2) Is it performant to only add neural concepts to existing rules, thus not allowing new rules to be made? Depending on the answers, the search space may become much larger and new heuristics have to be developed to approximate the performance without training each option. We have added a new discussion section to the text.

---

### Official Review · Reviewer_m7zW · 2025-10-27

**Soundness:** 2
**Presentation:** 2
**Contribution:** 4
**Rating:** 6
**Confidence:** 4

**Summary:**

This paper presents NeTheR, a neurosymbolic framework that automatically revises imperfect symbolic knowledge by introducing learned neural concepts. Unlike existing neurosymbolic systems that assume complete and correct background theories, NeTheR iteratively modifies a given logical circuit by inserting or removing neural concepts to improve predictive performance while preserving the symbolic structure. The method formalizes three types of circuit modifications (removal, conjunction, and disjunction with neural concepts) and uses a dynamically adjusted Sharpe ratio heuristic to estimate the performance gain and risk of each modification without fully training all candidates. Through iterative selection, compilation to arithmetic circuits, and end-to-end differentiable training, NeTheR efficiently identifies high-impact refinements. Experiments on nine multimodal benchmarks demonstrate that NeTheR consistently outperforms concept-based and neural baselines in both biased and unbiased settings, showing higher F1 scores and stronger robustness in low-data regimes.

**Strengths:**

The main strength of this paper is that it fills an important gap in the neurosymbolic learning literature by relaxing the common yet somewhat unrealistic assumption of perfect symbolic knowledge. Instead of treating background theories as fixed, NeTheR provides a principled way to revise and improve them through data-driven neural predicate invention. Although the proposed method seems relatively simple, this contribution is nonetheless meaningful and valuable.

More specifically, the method is conceptually clear and technically grounded, integrating symbolic structures with subsymbolic representations in a coherent manner. The use of a dynamic Sharpe ratio heuristic for modification selection is a sensible and interpretable design choice that balances performance improvement and reliability. The experiments are comprehensive, covering multiple modalities and settings, and consistently show that the proposed approach outperforms both symbolic and neural baselines.

**Weaknesses:**

The proposed framework is conceptually clear and practical, but the method itself appears somewhat procedural and heuristic-driven. The process of iteratively modifying a logic circuit and retraining is reasonable, yet the mechanism for inserting neural concepts through conjunction or disjunction could be discussed in greater theoretical depth. The dynamic Sharpe ratio heuristic, while intuitive and empirically effective, relies on heuristic estimations of best- and worst-case performance and on an empirically updated parameter $α$, without formal analysis of its behavior. Overall, the approach is sound and well-motivated but would benefit from stronger theoretical justification and more systematic evaluation of its heuristic components.

**Questions:**

Overall, I find the paper interesting but have several questions about the proposed method that may influence my final score if clarified:

1. How interpretable are the neural concepts invented during the revision process, and do they correspond to meaningful properties that can be understood or verified by humans, or are they merely latent constructs optimized for prediction?

2. To what extent does the revised theory differ semantically from the original one, and does this modification lead to genuinely improved reasoning quality rather than simply higher predictive accuracy?

3. How stable and repeatable are the learned revisions—if the algorithm is run multiple times with the same initial theory and dataset, will it converge to similar modifications or produce highly variable outcomes?

4. How does NeTheR ensure that the insertion of neural concepts does not compromise the logical consistency or interpretability of the resulting theory, especially when revisions accumulate over multiple iterations?

---

> ### Author Response · Authors · 2025-11-21
>
> >How interpretable are the neural concepts? Do they have real-world meaning?
>
> We kindly refer to the shared response on interpretability.
>
> >To what extent does the revised theory differ semantically from the original one, and does this modification lead to genuinely improved reasoning quality rather than simply higher predictive accuracy?
>
> We assume this question refers to the change in semantics by adding a neural concept to a propositional formula, thus introducing uncertainty in logic: instead of giving a Boolean prediction, the revised theory predicts a probability. This can improve the reasoning quality, as it is now able to express a more soft degree membership to the global concept, as is usually the case in NeSy/StarAI.
>
> >How stable and repeatable are the learned revisions—if the algorithm is run multiple times with the same initial theory and dataset, will it converge to similar modifications or produce highly variable outcomes?
>
> The approach will converge to similar modifications as long as $\alpha$ is similar. If the initial $\alpha$ is different, because through randomness a different set of modifications was chosen to determine it, then this may impact the estimated value of other modifications and the choices that are made. However, as $\alpha$ is updated iteratively, this effect will be limited.
>
> >How does NeTheR ensure that the insertion of neural concepts does not compromise the logical consistency or interpretability of the resulting theory, especially when revisions accumulate over multiple iterations?
>
> Please see our shared response on interpretability. If there are any remaining questions, feel free to let us know.

---

### Official Review · Reviewer_5svH · 2025-10-31

**Soundness:** 2
**Presentation:** 1
**Contribution:** 3
**Rating:** 2
**Confidence:** 4

**Summary:**

The authors tackle the problem of neurosymbolic learning under incomplete/imperfect knowledge. They use a DeepProbLog-like setup where leaves in an arithmetic circuit are neural networks. The new idea is that the method can create and insert new neural networks into the circuit to overcome its incompleteness.

**Strengths:**

- The task tackled is very challenging and important, and the idea of neural predicate invention is pretty novel
- Decently positioned in existing work
- Experimentally, the method works well on 9 datasets

**Weaknesses:**

- Unclear description of datasets and algorithms. In particular the generation of logic circuits to provide the classification task, and the  Sharpe ratio, which currently seems a bit ad hoc.
- Missing qualitative analysis of learned circuits/neural concepts
- If I understand correctly, the experimental setup is quite artificial, and the task may actually be quite easy. In practice it only invents 1.5 new neural concepts.

**Questions:**

Major clarification requests:
- A lot of neurosymbolic learning, like DeepProbLog, assumes no supervision on all features. Do I understand correctly that all relevant inputs are given during training? Eg, in the ATH dataset, the model gets the images and relevant attributes?
- The 'number of modification steps to obtain $M$ from $M'$' is not tractable to compute as there might be multiple paths to obtain it. This should be changed to just the number of updates, right? I don't think this distance metric is well-defined.
- Does every new neural concept get an entirely new neural network with the same inputs? What is to prevent the model from just learning an end-to-end classifier?
- Data generation process: Appendix B states that for each dataset, logic circuits $M^*$ are generated which provide the output label. This process is not sufficiently explained.
    - What are the features this logic circuit can use? Can they use also subsymbolic inputs (images)? In which case, what neural network is applied there?
        - If they also use subsymbolic inputs: does Nether use this neural network already at initialisation? (If so, that would make the task here quite easy)
    - How are these logic circuits generated? What is the size?
    - The authors remove nodes from the circuit $M^*$ to generate the incomplete circuit $M'$. How is this done? Especially the algorithm for without the structural bias?
- Please motivate the use of the critical difference diagram and cite the authors. I had to Google what these are, and they're the main experimental result.
- The Sharpe ratio part was very hard to follow and requires more details.
    - How is the expectation over $X_{nc}$ defined? How are $W$ and $B$ computed?
    - How is the risk-free rate computed? How is this best modification computed?
   - The derivation of equation 4 should be expanded (eg in the appendix).
- Some missing references of neurosymbolic methods with rule updating (no predicate invention, I think, or applied to theory revision).


Questions
- A common motivation of neurosymbolic is interpretability. What are the authors' thoughts on the interpretability of Nether?
- Are there any qualitative results? What do these circuits look like?

1. Daniele, Alessandro, et al. "Simple and Effective Transfer Learning for Neuro-Symbolic Integration." International Conference on Neural-Symbolic Learning and Reasoning. Cham: Springer Nature Switzerland, 2024.
2. Daniele, Alessandro, et al. "Deep symbolic learning: Discovering symbols and rules from perceptions." arXiv preprint arXiv:2208.11561 (2022).
3. Li, Zenan, et al. "Neuro-symbolic learning yielding logical constraints." Advances in Neural Information Processing Systems 36 (2023): 21635-21657.

---

> ### Author Response · Authors · 2025-11-21
> **Comments on the weaknesses (Part 1)**
>
> >Unclear description of (1) the generation of logic circuits; (2) the use of the Sharpe Ratio.
>
> We updated subsection C.2: Generating $M*$ and $M'$'' to clarify the generation procedure of the logic circuits. We also added two qualitative examples from the experimental evaluation of a ground truth circuit $M*$, a structurally unbiased and biased imperfect circuit $M'$ and their revisions in appendix D.
>
> The Sharpe Ratio is widely used in finance for risk-adjusted investment prediction, a problem that has parallels to our modification selection problem (lines 278-285). Its application is not ad hoc. To illustrate this better we have added Appendix B, an example of the modification selection that includes the computation of the worst and best case performance and the Sharpe ratio. If anything remains unclear, we are happy to expand an explanation.
>
> Steven E Pav. The Sharpe Ratio: Statistics and Applications. Chapman and Hall/CRC, 2021.
>
> >A qualitative analysis of learned circuits/neural concepts.
>
> We assume this refers to whether a learned neural concept corresponds to a human-understandable concept. We kindly refer to our shared response on interpretability.
>
> >The task may actually be quite easy.
>
> The task is hard, the existing competing methods all fail on multiple datasets: Appendix G, Table 5 and 6 show that each competitor has degraded performance ($< 0$) on multiple datasets.
>
> >Experimental setup is quite artificial.
>
> The experimental setup uses established datasets. Since these do not contain an initial imperfect model $M'$, we created one and introduced appropriate labels. To illustrate our method on existing labels (line 373), Appendix E describes an experiment on the original dataset labels. Table 4 shows the improvement of NeTheR in this setting.

---

> ### Author Response · Authors · 2025-11-21
> **Comments on Questions (Part 2)**
>
> >What data does NeTHeR use during training?
>
> The models receives all attributes present in the imperfect theory $M'$, and the images. The neural concepts in NeTheR only use the subsymbolic data and are trained using distant supervision. As a single neural network, NN only uses the subsymbolic data, while MLP+NN uses both symbolic and subsymbolic data (line 402-404).
>
> >The 'number of modification steps to obtain $M$ from $M'$' is not tractable to compute as there might be multiple paths to obtain it. This should be changed to just the number of updates, right? I don't think this distance metric is well-defined.
>
> In the problem description the distance $d(M, M')$ refers to the minimal number of changes to go from one to another. In practice, because this number is indeed not cheap to compute, NeTheR simply considers the number of modifications it has done so far. This may be an overestimate of the true distance. We have now clarified this in the text (lines 199; 318).
>
> >Does every new neural concept get an entirely new neural network with
> the same inputs? What is to prevent the model from just learning an end-
> to-end classifier
>
> Yes, each neural concept is a new neural network with the same input, which, in combination with the rest of the logical circuit, is trained end-to-end. The maximum distance $d_{max}$ prevents NeTheR from reducing input model $M'$ to a single neural network. However, it is technically possible to behave as a single neural network by adding a neural concept to a root disjunction, and eliminating the other logical branches by conjoining them with another neural concept. The performance would become as good as a single neural network. In practice, we find that NeTheR empirically performs much better than a single neural network (Figure 3, Table 5 and 6).
>
> >How are the logic circuits generated for the experimental evaluation?
>
> The logic circuit $M*$ only uses symbolic attributes (e.g., $age < 40$ or $temperature = High$). The initial imperfect logic circuit $M'$ is derived from this, also containing only symbolic attributes. More details on their construction process are now in subsection C.2 `Generating $M*$ and $M'$'. This also includes the size of the logic circuits. It the structurally unbiased experiment, $M'$ is a learned decision tree. To obtain $M'$, we use the training data and alter the parameters of the sklearn decision tree learner until $M'$ has an F$_1$-score smaller than $0.9$. In Appendix D, we added two qualitative examples from the empirical evaluation, of a ground truth circuit $M*$, a structurally unbiased and biased imperfect circuit $M'$ and of their revisions.
>
> >Please motivate the use of the critical difference diagram and cite the authors.
>
> We have added an explanation and reference in the caption of Figure 3.
>     A critical difference diagram, introduced by Demˇsar, Janez, and Benavoli et al. ($+16000$ citations), is a standard approach when comparing multiple methods across multiple datasets. It plots the average rank of each method (lower is better) and connects with an horizontal line the groups of methods whose performance is not significantly different. The latter is determined using the Wilcoxon's signed rank test.
>
> Janez Demˇsar (2006). “Statistical Comparisons of Classifiers over Multiple
> Data Sets”. In: J. Mach. Learn. Res. 7, pp. 1–30
>
> Alessio Benavoli, Giorgio Corani, and Francesca Mangili (2016). “Should
> we really use post-hoc tests based on mean-ranks?” In: J. Mach. Learn.
> Res. 17.1, pp. 152–161
>
> >The Sharpe ratio part was very hard to follow and requires more details.
>
> An example in Appendix B now shows the computation of $W$, $B$, and the resulting Sharpe Ratio. To go from Eq 3 to Eq 4, we assume that $X_{nc}$ is a two-point weighted random variable whose mean is $\mu=\alpha B + (1-\alpha)W$ and variance $Var(X_{nc})=\alpha(B-\mu)+(1-\alpha)(W-\mu)$ and use these as parameters for a shifted normal distribution. We have updated the text to make this more explicit (lines 305-307).
>
> >Missing references
>
> Thank you for drawing our attention to these works, they may prove very useful in the future. For the current submission, they do not fit well for the same reasons you have mentioned: they do not cover predicate invention nor theory revision.
>
> >What are the authors’ thoughts on the interpretability of Nether? Are there any qualitative results? What do these circuits look like?
>
> We kindly refer to the shared response on interpretability. Examples of circuits from the empirical evaluation are now in appendix D.

---

### Official Review · Reviewer_CmQ6 · 2025-10-31

**Soundness:** 3
**Presentation:** 2
**Contribution:** 2
**Rating:** 2
**Confidence:** 3

**Summary:**

This paper proposes NeTheR (Neurosymbolic Theory Revision) to address the common assumption in neurosymbolic AI that symbolic knowledge is perfect. NeTheR iteratively revises an initial, imperfect logical theory. Its key trick is leveraging predicate invention to introduce new neural concepts, which can learn from subsymbolic data (like images) to compensate for incomplete symbolic information. To guide this process, NeTheR uses a variant of the Sharpe ratio as a heuristic to identify and apply a limited number of high-impact modifications. The revised logic is compiled into a differentiable arithmetic circuit for end-to-end training. Empirical evaluations demonstrate that NeTheR outperforms its baseline competitors in this theory revision task.

**Strengths:**

- The paper deals with a core limitation of Neurosymbolic (NeSy) AI, which typically assumes perfect and complete symbolic knowledge. By focusing on revising imperfect theories, the work is more applicable to real-world, dynamic environments.
- Given the computationally impractical nature of training every possible modification, the paper proposes a novel heuristic based on a variant of the Sharpe ratio. This allows the model to efficiently identify high-impact modifications by estimating their risk-adjusted performance.
- The experimental results show that NeTheR clearly outperforms baselines.

**Weaknesses:**

- The NeTheR algorithm (Algorithm 1) relies on hyperparameter $d_{max}$. In the experimental setup, it is set to the number of systematic removals from M' to M*. However, in any real-world application, M* is not known, making it impossible to know how many removals were made to create. This method for setting $d_{max}$ is not practical, and the paper does not seem to offer a more reasonable alternative (such as setting it via cross-validation or using a performance-based early stopping criterion).
- I am not sure if the neural concepts found via the proposed performance-driven modification method actually have real-world meaning. Do they truly correspond to concepts that are interpretable to humans (like the paper's example of learning "whether an animal is white"), or do they just become another uninterpretable black-box feature that is simply "useful for classification"?

**Questions:**

- Why does the approach focus exclusively on introducing new neural concepts? The paper dismisses inserting single symbolic concepts ("insertion of a logical variable") as being "covered by traditional theory revision methods", but it does not justify why it avoids constructing new, complex symbolic concepts from logical combinations of existing symbolic variables.
- Does this preference for neural concepts imply that NeTheR's core value is solely in its ability to connect to subsymbolic data? If a task has no subsymbolic data (only symbolic data), is NeTheR's revision method still superior to traditional symbolic theory revision methods?

---

> ### Author Response · Authors · 2025-11-21
>
> > How can you know $d_{max}$ when $M^*$ is not known? The paper does not offer more reasonable alternatives such as a performance-based early stopping criterion.
>
> We stress that maximum distance $d_{max}$ is part of the problem setting, where it ensures that the learned model $M$ remains close to the original imperfect model $M'$. In practice, this value depends on the user's preference and the application domain.
>     It does not have to be equal to the true distance from a perfect model $M*$ (as $M*$ is not known). In fact, in the experiments without structural bias, most likely $d_{max} \neq d(M',M*)$. We argue this is practical; if the user is more lenient they can increase the value.
>
> In NeTheR, an overestimate of $d(M,M')$ is used as a stopping criteria (line 318) to not violate the problem setting constraint. This can indeed be extended, but not replaced!, with a performance-based early stopping criteria.
>
>
> >Do the neural concepts have real-world meaning?
>
> Please see the shared response on interpretability.
>
> >Why does the approach avoids constructing new, complex symbolic concepts from logical combinations of existing symbolic variables?
>
> Adding a combination of existing symbolic variables is covered also by the existing work on theory revision. We instead research specifically the optimal placement of neural concepts and the resulting end-to-end learning process.
>
> >Does this preference for neural concepts imply that NeTheR's core value is solely in its ability to connect to subsymbolic data?
>
> Correct, the key contribution is the ability to adjust an imperfect theory to also consider subsymbolic data.
>
> > If a task has no subsymbolic data, is NeTheR superior to traditional symbolic theory revision methods?
>
> NeTheR's focus is on subsymbolic data only. This does not limit expressivity, as one could apply a symbolic revision method and NeTheR sequentially (line 227). A more involved combination of both is an interesting avenue of research.

---

### Author Response · Authors · 2025-11-21
**Shared response**

We thank all reviewers for their feedback! In addition to the individual replies, we below clarify a shared point. We have also added this to a new discussion section in our submission. New additions to that submission have been coloured green for ease of reviewing.

**Interpretability and qualitative results**
 Neurosymbolic AI methods exploit existing knowledge to increase accuracy, robustness, with less data.
Interpretability also increases compared to purely neural models, because it enables the use of symbolic information and makes their combination symbolically understandable (i.e., and, or, negation). E.g., *swims=1* $\land$ *neural\_concept* is more interpretable than only *neural_concept*.

NeTheR introduces neural concepts that do not have a prior intended meaning (line 150). This trades-off interpretability for accuracy compared to a fully symbolic initial model $M'$. However, not having a prior intended meaning has only limited impact on interpretability in the context of neurosymbolic AI: a neural network does not necessarily align with an intended concept such that an analysis (for example using saliency maps etc.) is required for a robust explanation regardless of having an intended meaning.

Furthermore, because there is no intended meaning it is impossible to provide a qualitative assessment of whether the learned concept matches an intended one. Even in the experiment with a structural bias, where symbolic information is removed: it would be incorrect to only compare the learned neural network to the removed information because it is possible to insert elsewhere correlated concepts that amount to the same semantic model.

---

### Meta-Review · Area_Chair_QYMa · 2026-01-07

**Summary:**

The reviewers raised concerns regarding the inappropriate experimental setup, insufficient discussion, limited qualitative analysis, and restricted theoretical depth. Most of these concerns remain unresolved after the rebuttal. Therefore, I recommend rejecting this paper.

**Reviewer Concerns:**

- Reviewer CmQ6: Inappropriate experimental setup, unclear concept interpretability
- Reviewer 5svH: Unclear description, missing qualitative analysis, Inappropriate experimental setup
- Reviewer m7zW: limited theoretical depth
- Reviewer KUoG: limited scalability, insufficient theoretical guarantees

**Reviewer Scores:**

- Reviewer CmQ6: No
- Reviewer 5svH: No
- Reviewer m7zW: No
- Reviewer KUoG: No

---

### Decision · Program_Chairs · 2026-01-26

Reject